# Trust, Connection and Equity: Can Understanding Context Help to Establish Successful Campus Community Gardens?

**DOI:** 10.3390/ijerph17207476

**Published:** 2020-10-14

**Authors:** Pauline Marsh, Suzanne Mallick, Emily Flies, Penelope Jones, Sue Pearson, Iain Koolhof, Jason Byrne, Dave Kendal

**Affiliations:** 1Centre for Rural Health, College of Health and Medicine, University of Tasmania, Hobart 7000, Australia; Suzanne.Mallick@utas.edu.au; 2School of Natural Sciences, University of Tasmania, Hobart 7000, Australia; Emily.Flies@utas.edu.au; 3Menzies Institute for Medical Research, University of Tasmania, Hobart 7000, Australia; Penelope.Jones@utas.edu.au; 4School of Medicine, College of Health and Medicine, University of Tasmania, Hobart 7000, Australia; Sue.Pearson@utas.edu.au (S.P.); koolhofi@utas.edu.au (I.K.); 5School of Geography, Planning and Spatial Sciences, University of Tasmania, Hobart 7000, Australia; Jason.Byrne@utas.edu.au (J.B.); Dave.Kendal@utas.edu.au (D.K.)

**Keywords:** campus community garden, health, socio-spatial connection, trust, sustainability, university students, wellbeing

## Abstract

Campus community gardens (CCGs) can potentially improve student health and wellbeing, mitigate social and ecological problems, and nurture university-community relationships. However, CCGs are located in complex socio-political and ecological settings and many community gardens struggle or fail. However, few studies have assessed the socio-political/ecological context of a garden setting prior to its development to understand the potential barriers and enablers of success. Our study assessed the socio-spatial context of a proposed CCG at a student university accommodation site. We engaged diverse university and community stakeholders through interviews, focus groups and a survey to explore their perceptions of the space generally and the proposed garden specifically. Visual observations and public life surveying were used to determine patterns of behavior. Results confirmed known problems associated with an underutilized site that provides little opportunity for lingering or contact with nature; and unknown barriers, including socially disconnected stakeholders and community distrust of the university. The research also uncovered positive enablers, such as stakeholder appreciation of the social, wellbeing and ecological benefits that a CCG could deliver. Our findings suggest that an in-depth exploration of a proposed garden context can be an important enabler of its success.

## 1. Introduction

People’s disconnection from Nature is increasingly recognized as harmful to wellbeing and the natural environment. Researchers, activists and policy makers are becoming more interested in interventions that can generate (re)connections between people and nature. Campus community gardens (CCGs), sometimes called student-led food gardens, are an example of such an intervention. As sites of applied learning and research, they are often embedded into curricula, including horticulture, landscape design and sustainability studies [1]. CCG research and teaching benefits are recognized across diverse university departments, including the natural sciences, arts, social sciences and geography [2] and CCGs are also playing an important role in university sustainability strategies, as part of the global ecological sustainability agenda. They can foster a heightened sense of responsibility for the environment [3,4] and engage the broader community with issues such as food security [5] energy use, mental health and ecological integrity [6].

For instance, Baker and Bilbro [7] have called for CCGs to be used for more than the utilitarian acquisition and application of student knowledge, suggesting they can shape imagination, build gratitude and enable students to care deeply about place and health. Such gardens can also improve academic success among school children [8], and foster engagement between the broader community and a university when they are established as sites for activities like citizen science and community events [1], social and creative-arts events and other projects [9]. CCGs can bolster connectivity between multiple actors—students, staff and the wider community—and between individuals and natural environments [4,8,10]. By participating in gardening, students and community members can experience improved physical and psycho-social wellbeing [11,12] as well as stronger social connections, a deeper sense of place, and a sense of stewardship of the natural environment [7,12,13]. This multi-faceted connectivity is at the heart of the wellbeing, socio-spatial and ecological potential of CCGs.

Although planting vegetables together in communal campus sites to improve human and planetary wellbeing sounds simple and attractive, establishing and maintaining a CCG requires considerable planning, coordination and investment of time and energy. For these reasons, despite best intentions, some CCGs can struggle to survive and others fail outright [3,5,14]. Yet we know comparatively little about the role that diverse pre-occupancy drivers, barriers and enablers play in CCG success. “Youth” gardening is under-researched [15] and comparatively little is known about university student access to community gardens [9] or indeed about student engagement in green spaces in general [16]. Sometimes, discrepancies exist between institutional aims and processes, and the ideas of students or the wider community - and these differences can be problematic [17]. Perhaps most significantly, university campuses are not simple environments to understand; they are highly complex socio-political and ecological settings [3].

This paper reports the results of a pre-occupancy study of a campus community garden site in Hobart, Tasmania. The study arose from a desire by the authors and university management, who funded the project, to ensure that the CCG was established successfully to deliver positive health, wellbeing and ecological outcomes. By uncovering the complexities of the context for the proposed garden, especially the drivers, barriers and enablers of garden success, we aim to positively influence the design and ongoing management of similar gardens in Australia and internationally. We begin by overviewing the academic literature on campus community gardens. Next, we present the study site and the methods used. We then discuss our findings, highlighting what they reveal about the importance of understanding the broad context. We conclude with a suggested strategy for the effective establishment of CCGs and point to directions for future research on the socio-spatial, wellbeing and ecological benefits of these gardens.

## 2. The Role and Function of Campus Community Gardens

### 2.1. CCGs and Physical, Psycho-Social Wellbeing

Community garden-based research demonstrates positive impacts of community gardens on physical, mental and cognitive health. Benefits are attributed to the core garden constituents of fresh air, social interaction, healthy food, reduced stress and physical exercise [18,19,20,21,22,23]. Closely associated are benefits owing to nature’s ability to facilitate recovery from mental fatigue [24], reduce risk of psychological distress and build healthier cortisol profiles [25]. Oliver Sacks in an essay on hortophilia (the desire to interact with, manage and tend nature that is deeply instilled in humans) suggests that gardens might be “*More powerful than medicine*” ([26], p. 245). Community gardening, in general, can facilitate wellbeing through social change and improved social equity from the opportunities it affords for civic engagement and political action [11,27,28,29].

Universities, especially those located within cities, can and do play a role in increasing the liveliness of public space. Cilliers et al. [30] note that public spaces that are underutilized can be enlivened through the presence of education institutions, coupled with design that enhances and supports the natural movement and flow of people through urban space. Universities also generate impacts on the environments that surround them—such as increased vehicular traffic, resource consumption, pollution and increased property values [31]. Community gardens located on university campuses (hereafter campus community gardens (CCGs)), can mitigate some of these impacts, as well as conferring a range of benefits on students and the broader community. They thus play an important role in a university’s social license to operate [32].

CCGs have been used to deliver onsite, outdoor student wellbeing programs [16] and host creative and social events to facilitate student engagement and maximize the potential for enhanced mental wellbeing. The high risk of mental ill health for the student cohort [33] that is associated with academic stress [34] means they have much to gain from community gardening. Contact with nature among high school students contributes to perceived restoration [35]. Similarly, a study with university students found the presence of green spaces on campus allowed students to recover from cognitive fatigue [16]. CCGs can facilitate social connections for people from minority population groups, including international students, who are more at risk of social isolation [36]. For example, Holt [16] found frequent, active interactions in university green spaces by students led to higher quality of life and increased feelings of happiness. Baur [12] also found improved wellbeing was linked to volunteer engagement in a CCG.

### 2.2. CCGs and Connection to Nature

Built up areas, including university campuses, provide limited options for connecting with nature. In some cases, campus green spaces may be the only accessible natural space for students [12]. Parks, streetscapes, gardens and informal green spaces can help develop and maintain human connections to nature [37] as can community gardens [38]. Not only is this contact important for health, but also student success and completion rates [8,12]. Researchers have recommended university management support the establishment and maintenance of natural spaces on campus [12] both for active recreation and for solitude, rest and restoration [16,39]. Myers [38] notes that community gardens are particularly suitable form of urban nature to provide everyday, incidental and fluid contact options. Not only, she argues, can they be created so people can engage with nature in all types of weather conditions, and offer variety in the scale of agriculture or cultivation, they can encourage people to linger and thus enhance the contact experience.

Further, student accommodation sites are generally home to students who are new to a university and potentially to city living too. While demand-driven initiatives at Australian Universities (to increase student numbers from previously under-represented population groups) have increased the numbers of enrolments from people living in rural and remote areas in Australia, overall drop-out rates remain higher for regional or remote students than their urban peers (by age 23) [40]. There are many reasons for incompletions, one being that moving from a rural area into the city necessitates not only a physical transformation, but also a psychological reimagining that can be dogged by self-doubt and uncertainty [41]. Fleming and Grace [40] found some rural students had not seen a university campus or visited a city prior to enrolment. Schultz [42] observed disconnectedness, lack of solitude and intermittent desires to get away from the city among first year university students. CCGs can thus function as a type of therapeutic landscape, as well as having broader sustainability benefits.

### 2.3. CCGs and Ecological Sustainability

Contemporary urban farming and agro-ecology approaches encompass principles of increased nature connectedness and improved ecological sustainability [43]. Community gardens regularly accommodate sustainability practices, such as composting and recycling waste, and have been referred to, eloquently, as sites of embodied sustainability [44]. They contribute to sustainable ecological and urban development, generating multiple environmental benefits [45,46] including pro-sustainability attitudes [21] and sustainable food systems—in keeping with the ‘Slow City’ agenda [47]. Australian and US community gardens and UK allotments are regularly established in communities of low socio-economic status, and often on parcels of ‘waste’ land that are reclaimed for food production. Some UK researchers argue that small scale allotments can grow sufficient produce to meet daily fruit and vegetable consumption requirements [48]. More recently, community gardening has been recognized as important in mitigating the impacts of climate change [49] and enhancing biodiversity [50].

Universities are well-placed to be leaders in ecological sustainability on local and global levels [51]. Colding and Barthel [52] advocate for universities to actively participate in the global sustainability agenda by implementing measures to reconnect students with natural areas on campus. Reconnection, they argue, can be facilitated through learning opportunities and applied research projects. However, they also suggest retrofitting campuses in line with biophilic design principles to support ecological and socio-spatial connectivity. Duram and Kleine [53] argue similarly, they found that CCGs can increase knowledge of sustainability, as well as institutional sustainability through a range of measures. Campus gardens play a role in Colding and Barthel’s [51] connectivity agenda: stakeholder stewardship and the public prominence of the CCG are considered beneficial for developing a stronger sense of place as well as pro-environmental behaviors.

### 2.4. Key Factors for CCG Success or Failure

While there is little academic literature that evaluates the success or failure of CCGs, Duram and Williams [2] suggest key elements for a successful CCG, including: Long term funding, productive gardens, trained workforce, a visible presence, hands-on learning, and links with broader sustainability networks. To this list, we add strong student participation and institutional support [53]. Stephens et al. [51] found a clear organizational structure, close ties to the university curriculum and management, a strategic plan and innovative fundraising events were instrumental in CCG success. Understanding volunteer engagement enablers [9,54] is also important, as is stakeholder support from the broader community and university affiliates. Indirectly, Anderson and colleagues [9] found that personal motivations and regular and interactive communication were key components to securing external financial support.

Stephens et al. [51] have identified the barriers to CCG success as including: Competing priorities and obligations; issues around modes of communication and lack of information; inconvenient/poor timing of the organized activities and perceived lack of opportunities for involvement. CCGs can fail when design is inadequate, people are not engaged and university management is not committed (financially and otherwise) to their success [14]. Other barriers include when student turnover is high, there is a lack of guidance and limited financial resourcing [5] and an overreliance on volunteerism from a transient student community [21]. Universities aspire to be connected with the communities in which they are located, to be ‘outward looking’ [52] and leaders in sustainability [2]. However, Colding and Barthel [51] suggest that university boards can sometimes function as a barrier to a sustainability agenda, depending on the alignment with broader fiscal, value and educational demands. Increasingly, universities struggle to gain the confidence and trust of the public, and power imbalances between the institutions and community members are common [55,56]. CCGs offer one way that universities can improve their social license to operate.

## 3. Methods

This study employed a mixed-methods approach. We characterized the socio-spatial and wellbeing context of the pre-garden site primarily through visual observation and engagement with a range of stakeholders. Through discussion, we sought people’s perceptions of the space, their thoughts about a CCG and potential constraints. We gathered input from those most impacted by the planned establishment of a CCG - students, people currently using the public spaces associated with the accommodation facility, the surrounding retail community and gardeners from nearby urban community gardens (hereafter, stakeholders). Informed by a critical geography methodology [57] and a desire to understanding, know and experience the world via multiple perspectives a mixed methods design was implemented [58] using semi-structured, in-depth interviews, a focus group, an online survey, public life observation, and video observation. For a full summary of methods and relevant stakeholder codes, see Table 1. Ethics approval was obtained from the Social Science Human Research Ethics Committee at the home institution (Reference H0018320).

### Site Description

In 2019, the University of Tasmania (UTAS) commenced planning for a community garden at one of its recently built inner-city student accommodation sites, the Hobart Apartments. UTAS is the single university in Tasmania, an island state with a profile typical of regional areas in Australia: a small population (state population: ~517,000, spread across cities and regional towns. The primary UTAS campus is located just outside the regional capital city, Hobart (population approximately 250,000) with several satellite campuses in other regional towns), widespread social disadvantage and poor health outcomes. In Tasmania, living in a rural location during Year 12 correlates with decreased chances of ongoing study five year post high school completion [40]. According to the Tasmanian Chamber of Commerce: “*Tasmanian’s … greatest challenges and potential still exists in the low levels of education, literacy and health outcomes of our population*” [59]. Offsetting this disadvantage is a potential health and wellbeing advantage; regional areas are in close proximity to sometimes extraordinary wilderness and natural environments. The results from Australia’s Social Progress Index [60] in 2019 captures this scenario well: for the Health and Wellness category Tasmania ranks seventh of the eight states and territories, while in the Environmental Quality category Tasmania ranks first.

In 2019, UTAS announced its intention to consolidate its southern-based campuses in the city over the next 10–15 years. This intention elicited a mixed response. UTAS staff and students raised concerns about possible negative health and wellbeing repercussions, associated with an anticipated worsening of social and nature disconnections. The broader public were concerned about the increased physical presence of UTAS buildings, staff and students in the city (Figure 1). One high profile UTAS alumnus challenged UTAS to provide “*solid proof that each* [new] *building will be to the benefit and not detriment of Hobartians’ daily lives and the amenity of the city*” [61].

The Hobart Apartments are located in the Midtown retail precinct of the state’s capital, Hobart. It is one of several new student accommodation buildings UTAS established in the city during 2017–19. The UTAS Hobart Apartments site is located on a corner block and extends behind existing heritage shops and buildings. It consists of 430 apartments with a University shop front, a cafe and a car park (with public parking spaces). The Hobart Apartments occupy 12,238 m^2^, and the outdoor areas are a mix of public and student-only spaces, in and around the three wings of residential apartments (Figure 2, Figure 3, Figure 4 and Figure 5).

The CCG was proposed to extend across all of the outdoor spaces, including the public and private sites.

## 4. Data Collection

### 4.1. Interviews and Focus Groups

Local surrounding businesses and people moving through the site were invited to participate in semi-structured interviews. With consent, they were asked for their opinions on: the current use, appeal and function of the site; the health and wellbeing impacts of green spaces and community gardens in general; the proposed UTAS move to the city and ideas for possible usage of the site. A focus group was also held with participants experienced in community gardening, urban farming, or inner-city greening. Through open-ended questions, the focus group participants discussed the potential of a community garden at the site and the associated challenges and benefits. All interviews and the focus group discussion were digitally recorded and transcribed.

### 4.2. Site Observation

Non-participatory on-site observation was also undertaken by researchers at designated intervals to gather visual information about current movement through the space and the activities that people undertook there. Using modified public life survey tools [62] to survey for liveliness, we combined the people moving count and stationary activity mapping to record numbers of people moving through the space and at various times (movement through), as well as the frequency and type of activities undertaken (stationary activities). The surveyor followed a regular circuit through the upper and lower sections of the site and the survey sheet captured time, weather and observation notes.

In addition, a video camera was located in two prominent spots, alongside a sign alerting passers-by to the recording, at additional designated times. Data collection took place over a few weeks during early spring at the time of changeable weather. The footage further captured movement of people in and through the space. Combining the upper and lower sites we obtained 7 hours of video footage and 320 minutes of surveying (physical observation) collected over five days at various times of the day during the week and on the weekend (Table 2).

### 4.3. Online Questionnaire

A short online questionnaire was developed to provide some insight into current quality of life, connection to nature and trust in the university by local communities (particularly the student body). Participants were recruited through posts on residential student Facebook groups and posted notices at student accommodation. Movie vouchers were offered as incentive and informed consent was implied by completion of the survey. The survey instrument included several standard psychometric Likert-type scales measuring both wellbeing via the Personal Wellbeing Index [63,64] and connection to nature via the 6-item Nature Relatedness [65]. A scale measuring trust in institutions procedural fairness in engaging the community was adapted from a published scale [66] and separately phrased for students and local community members. Additional data were collected on student enrolment (degree and year of enrolment), aspirations (intention to finish their degree) and basic demographic information. An open-ended question was used to collect additional qualitative data on participants’ views on the establishment of a community garden in the city as part of the campus move.

## 5. Analysis and Results

A convergent analysis approach was applied, using the conventions appropriate to each method. In this section we explain the analysis process and present the results for each method.

The interviews focus group data and open-ended survey question were analyzed using reflexive thematic analysis techniques and inductive coding [67,68]. Two researchers initially read each of the transcripts independently and assigned relevant codes. The codes were then synthesized through a dialogical process of discussion, reflection and revision. This process generated themes and subthemes which were subsequently discussed, reflected upon and refined by the wider research team. The final four themes pertaining to the potential impacts of the campus community garden were health and wellbeing impacts, social connectivity, nature connectedness and Midtown community building (Appendix A).

The observational data and video footage were analyzed through a similar interpretative process involving viewing, coding (patterns of movement, pace, and interactions), reflection and discussion by two researchers until the patterns of human interaction with and within the space were determined. Two key findings are generated by analysis of the observation data. Firstly, there is far greater continuous movement though the space than lingering (Figure 6) and secondly, warmer weather encouraged lingering, but generally activity was intermittent and short in duration (Table 2).

The weather during the week of observation was typical of a Tasmanian spring with a dynamic mix of cold weather, intermittent sunshine, cloud cover and wind. Parts of the site were variously in sun and shade as the day passed, and some areas were subject to strong wind depending on the direction. In this regard, patterns of behavior were predictable. When the weather was either ‘sunny’, ‘part sun’ or ‘warm’ episodes of stationary activities increased. In cooler times people stood or sat in the sun, while on sunny days they waited in the shade. In rain and wind, no people sat in the open spaces, and when the undercover areas were in full shade (outside the café) no one sat in them. The greatest number of stationary activities took place on Friday afternoon and Saturday (Table 2) at which time the space was occupied by a group of parkour enthusiasts in which people made the most of the concrete features of the lower space. Other activities comprised such things as eating lunch, talking on the phone and waiting with children (Figure 7). Apart from a couple of students who studied at a table, most of the stationary activities lasted only a few minutes. Despite “the Loop” (see Figure 5 above) running continuously for the 740 min of observation, only 4 people were observed to stand and watch it for a few minutes.

A total of 64 respondents completed the survey. This included 43 current students (23 living at the Hobart Apartments and 14 living at other student residences in Hobart), and 21 community members. Fourteen students identified as being from rural or regional Tasmania before studying, 15 from mainland Australia and six international students. Fifty-nine percent of respondents identified as female, 37 percent as male, and 4 percent as non-binary. For quantitative survey data, we first validated the psychometric properties of the revised trust scales by calculating Cronbach’s alpha on the student and community versions of the procedural fairness trust scale. These showed that the scales had a high level of internal validity (standardized Cronbach alpha = 0.97 and 0.99 respectively). Means and 95 percent confidence intervals were then calculated for the Personal Wellbeing Index, nature relatedness and trust scales. These were compared with results from previous studies [66,69] plus unpublished data on nature relatedness from a contemporaneous national study of 2000 Australians [70].

A comparison of average levels of personal wellbeing, nature relatedness and trust with the results of other studies shows that survey respondents had relatively low levels of personal wellbeing and high levels of nature relatedness (Figure 8). Levels of institutional trust were analyzed separately for students and the general community. While the small sample size increased uncertainty (as indicated by the large error bars), there was some indication that levels of trust were relatively low in both the student population and the general community.

Examining each individual Personal Wellbeing domain shows that respondents had lower levels of ‘future security’, ‘feeling part of community’, ‘achievements in life’, ‘health’ and ‘life as a whole’ (Figure 9).

In addition to the above, 24 participants provided a brief response to the optional open-ended comments section of the survey. Three broad themes, related to our aim, were generated from our analysis of these: green space, community space and living space.

Critical interpretation of these results provides in-depth insight into the local context for this campus community garden and identifies likely enablers and constrains for its capacity to have positive impacts on people and natural environments. In the following section we discuss what the results collectively reveal about the pre-garden socio-spatial and ecological context and the anticipated constraints for the CCG.

## 6. The Pre-Garden Wellbeing, Socio-Spatial and Ecological Context

### 6.1. Health and Wellbeing

There was widespread acknowledgement of the health and wellbeing benefits of green spaces, nature and gardening across all stakeholders in our research, and people drew upon both personal experience and empirical evidence to make this point. Two factors appeared to augment the perceived importance of the greening intervention in this context: the inner-city location of the site and that the site was for students to live in. Student mental wellbeing was perceived as an important outcome of improving the greenery of the area, and the establishment of a CCG, for many community members. The retailer’s prediction of a looming student mental health crisis from disconnection to nature, for example, was expressed in various ways by many others. The importance of mental health as a factor was reinforced by survey respondents reporting relatively low levels of personal wellbeing.

The perception of potential health and wellbeing benefits also extended to the health of the broader community. Access to green spaces for people living and working in a city was seen as desirable because they were calming, especially if away from traffic noise and buildings. Some wanted quiet areas for rest and restoration, others thought the sites should be greened to enable more active recreation. The perceived effects on health may have been emphasized due the research being undertaken at onset of spring, when people were appreciating the opportunities to be outdoors in finer weather after a Tasmanian long and dark winter.

However, the pre-garden site was failing to deliver positive health or wellbeing benefits to either students or the community. Students stayed in their rooms, people walked through the site without lingering and many sought out restorative greenery in other areas of the city, or beyond. Stakeholders who grew up, or currently lived, in rural Tasmania did not use the site, or the city in general, as a means of connecting with nature. The results show a need to change the physical structure of the site to enable better contact with nature, as well as to find measures to encourage student and community participation in the site. Despite all of this, the CCG was considered a welcome intervention having the potential to improve psycho-social human health and wellbeing.

### 6.2. Aspirations for Social Connectivity

Most stakeholders—students, public and local traders—were using the site to transit from an adjoining street, the onsite-car park or the student accommodation, further into the city. People were not inclined to linger in the site and regular passers-by confirmed it was often empty. One student described how he had chosen the Hobart Apartments because he thought they would provide ‘a community’ but felt this had not happened. This social disconnection was in part attributed to the physically unwelcoming site. One person thought the site was a ‘memorial’ for a tragic event, others were less extreme but expressed a similar sentiment.

There was also a strong desire to use the outdoor space in ways that improved opportunities for social interactions and community building. Some stakeholders saw the UTAS presence in the Midtown region as an opportunity to continue building the vibrancy in the area. One business noted that as the Hobart Apartments site was being built the vacant shops started to fill and several businesses were optimistic about the future plan to consolidate more of UTAS to the city. The community garden was considered to have positive potential to further increase community participation and engagement.

The reality of the unwelcoming space contrasts with the ambitions for the space held by surrounding businesses. Midtown is characterized by small locally owned retail and service outlets, some run by families, who consider themselves the creative and energetic edge of Hobart city. They are enthusiastic place-makers. Although the site was not considered ‘activated’ or to be engaging for either the community or the students, there was much goodwill toward changing this and towards utilising the space to bring resident students and other community members together. The car park, café area and open spaces were thought good ‘enablers’ for events, activities and meeting up with people, and people recalled a successful community event held soon after the Hobart Apartments opened.

The Midtown community of traders were committed to revitalizing the city in creative and innovative ways, and felt it was important that students weren’t physically or mentally ‘separated out’, but engaged in this broader trading community. One local organization was already interacting with students and saw enhanced opportunities to work more with UTAS. A CCG has the potential to be consistent with, and complementary to, the Midtown community’s sense and vision of place, as well as stakeholder ideas about the value of city greenspace.

### 6.3. A Place in the Global Ecological Sustainability Agenda

When informed about the possibility of a campus community garden on the Hobart Apartments site, students and other stakeholders were enthusiastic about the opportunity it would give them to be engaging in nature and in broader issues of ecological sustainability. Although some participants appreciated the ‘industrial’, ‘modern’, or ‘concrete-blocks-like-art-installation’ appearance of the space, the improvements they suggested were overwhelmingly concerned with additional greening or including natural materials.

Participants valued green spaces and considered access to them to be what made the University of Tasmania unique. This contributed to individual decisions to study at UTAS, as it was thought to provide a relaxing and ‘eco-friendly’ environment. However, these opinions relate to the existing, suburban UTAS Sandy Bay campus, which is surrounded by bushland and has urban green spaces permeating the campus. Participants viewed the move to the city as one that would detract from this appeal and felt there were insufficient green spaces planned and missed opportunities to establish a green “wilderness” in the city. Further, survey respondents had relatively high levels of Nature Relatedness, suggesting a need for opportunities for students and the local community to connect to nature.

The fact that several local traders suggested the university collaborate with them to introduce shared sustainable practices on the site indicates a sense of collective responsibility, beholden to both an inner-city community and a university community. The focus group considered the potential community garden to have educational value for students and community members alike to learn about sustainability measures such as waste management, biodiversity and composting. The possibility that UTAS might steer a collaborative ‘hub’ for community gardens in the area was encouraged by focus group members. However, some stakeholders suggested there was a need for deliberate strategies to ensure that students connected with nature in the garden, that is, they assumed that students would be disinclined to do this of their own accord. The transient nature of volunteers and students, depending on the time of year, was also raised, as were other commitments on the university calendar such as academic assessment times.

There was strong interest in establishing a CCG as a means of raising awareness of and improving the ecological sustainability of Hobart city and globally. Participants valued connection to wilderness as well as ecological sustainability. Across stakeholder groups people expressed a sense of personal responsibility for addressing climate change and increasing biodiversity. Focus group participants were particularly outspoken on this issue, many of whom are part of a dynamic sustainable community gardening and urban greening movement in Tasmania. They suggested the CCG model have sustainability at its core rather than a focus on large-scale food production which was considered unachievable on the site.

Focus group participants also stressed the importance of planning, sticking to a ‘model’, and including professional horticultural management and/or a facilitator to ensure the success of a community garden. Implementation of a community garden may require a multi-staged approach, but participants saw an opportunity to make the garden a place for interaction and collaboration from other urban agroecology projects and gardening entities in the city.

## 7. Anticipated Constraints

### 7.1. Socio-Spatial Disconnections

At the time of our pre-occupancy research, the site was not facilitating people’s connection to nature or to each other. The site was underutilized on all days and periods of our observation, apart from the visiting parkour crowd on a weekend. Most people were taking a short cut through rather than stopping and spending time; the effect was the creation of outdoor corridors, rather than spaces for socializing or communal recreation. The through movement was in part influenced by weather conditions and cool microclimates, which did not encourage sitting or standing still for long. In rain and wind, no people sat in the open spaces, and when the undercover areas were in full shade (outside the café) no one sat in them. However, these climatic factors were exacerbated by physical factors, such as extensive shade, cold concrete blocks and few wind breaks. People made suggestions that indicate a desire for a more comfortable and restorative outdoor space: chairs, grass to lie on, a children’s slide, bean bags, couches, a fountain or water feature.

One major barrier to the use of the outdoor areas was that people were confused as to whether it was public or private space, including the university staff managing the student accommodation site. This was exacerbated by a diversity of stakeholders with different ideas about public/private sites. Concerns about outside persons accessing certain areas on the university campus late at night had negative impacts on survey respondents’ sense of security and personal safety. A person who was asked to move on from the public BBQ, for example, was homeless at the time and had been cooking dinner. He did not know why the accommodation manager moved him on, but perhaps the manager was unclear about the private/public space delineations and felt the site should be used for other activities, or that a homeless person was not welcome at the student accommodation.

Part of the challenge for place-making and liveliness in spaces where the public and private intersect is meeting these various and varied needs. This is perhaps heightened in a regional island setting with a small population, where historically home spaces and the city center have been separated. Mixing private residences alongside retail outlets is relatively new for Hobart.

### 7.2. Distrust

Although UTAS hopes the planned move to the city creates “*a campus that is welcoming to all and that builds community*” [71] our results show the university is perceived as irresponsible with money, particularly in light of the broader context of disadvantage in the state. While there was evidence of a historical culture of respect for the university, that has been challenged by more recent decisions and actions in the city. Some raised concerns that the university was catering to international students rather than local Tasmanians. Locally, the actions of UTAS buying property and developing large buildings in the city was interpreted as a display of arrogance and disrespect for community values. There were recollections of broken promises—the university had said they would engage the local community in decisions around use of this space but had not followed through.

Many retail business participants claimed there was little direct engagement between students and retailers. Students rarely visit the Midtown businesses and businesses are not regularly involved in the student accommodation sites. On the whole, there seems to be a social disconnection between UTAS, the student residents and the surrounding business community. Business participants felt it was critical that UTAS engage directly with them in the ongoing development processes at the site. Our survey suggests students and the local community have relatively low levels of trust in UTAS to involve students and the community in decision making.

Trust in the university could best be described as fragile. Although the Hobart Apartments were established under a scheme designed to provide affordable accommodation (the National Rental Affordability Scheme), stakeholders were concerned that the university’s continued move into the city would inflate rental and accommodation prices and make inner-city living less affordable. For many, the lack of student patrons and customers was one of several promises that UTAS had made but not delivered on. Other issues related to communication, consultation and participation. People wanted the outdoor spaces to welcome the broader community and not be purely student-focused, and they emphasized the importance of talking with the community and (re)building the relationship with community into the future.

This research reveals deep concerns about UTAS role in perpetuating social and economic inequities in Tasmania. Distrust in the institution will constrain community and student engagement in further initiatives planned by UTAS in this space, including the CCG. It also undermines the University’s social license to operate. There was, however, evidence of goodwill towards improving relationship between the Midtown community and the University that may serve as a counterbalance. Creating opportunities in the city were considered important for the greater role this could play in shaping the future of the island.

### 7.3. Research Limitations and Future Direction

The observation methods used in our study cannot reliably reveal whether one episode of movement or activity relates to a single person or multiple visits by the same person (hence the number or a measure of activity are reflected rather than number of individual people utilizing the site). The researcher observation sessions were undertaken by a single researcher moving on foot through the two spaces, in a regular pattern. While the survey responses were consistent with qualitative findings, the small sample size of the online survey means that care should be taken in extrapolating results to the whole student body or local community.

Since this research was undertaken, the COVID 19 pandemic has changed the way people engage with public spaces. This, coupled with our results and the work of others, points to future directions for Campus Community Garden based research, exploring the following questions:How have public health restrictions impacted CCGs and their capacity to impact positively on health, wellbeing and social connectivity?What are the roles and impacts of CCGs in regional areas of low socio-economic status but with easy access to nature?What are the variations in experiences of CCGs for discrete student and population groups, such as international students and domestic students?What benefits nature and greenspace from the establishment and maintenance of CCGs?How do CCGs impact on university and course selection and satisfaction, and how might universities maximize this?

In addition, there is a dearth of research into the long-term sustainability of CCGs, and therefore little knowledge about the factors that directly impact on the success or failure of gardens to have positive impacts.

## 8. Conclusion: A Strategy for Creating an Effective Campus Community Garden

We conclude by proposing a strategy for effective campus community gardens that considers input from existing evidence alongside particularities of a local context (Figure 10). Our aim is not to suggest there is a one-size-fits-all model for successful CCGs, but rather to provide a guide for creating CCGs so that they can best achieve their manifold functions: providing an educational, social and community hub that connects people with each other and with nature; improving physical and psycho-social health and wellbeing; enhancing ecological sustainability; driving social equity in regions of disadvantage; and restoring trust between universities and the communities in which they are located.

This research demonstrates the value of gaining an in-depth understanding of a pre-garden context before a campus community garden is established. Without this understanding, potential significant barriers and enablers will go unnoticed, and opportunities to ensure the garden is successful and as impactful as possible can be missed.

## Figures and Tables

**Figure 1 ijerph-17-07476-f001:**
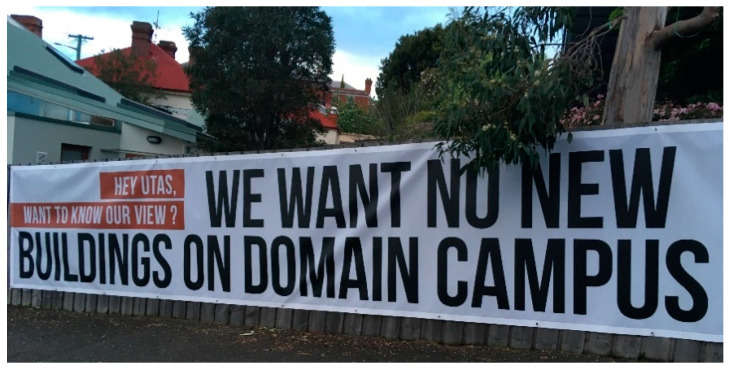
Inner-city residents respond to the Southern Future Strategy announcement.

**Figure 2 ijerph-17-07476-f002:**
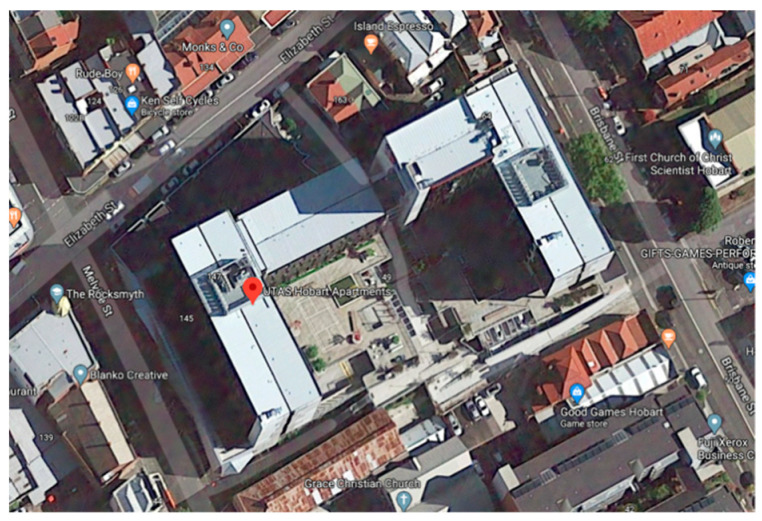
Aerial view of the student apartment complex and proposed garden sites [Google Maps].

**Figure 3 ijerph-17-07476-f003:**
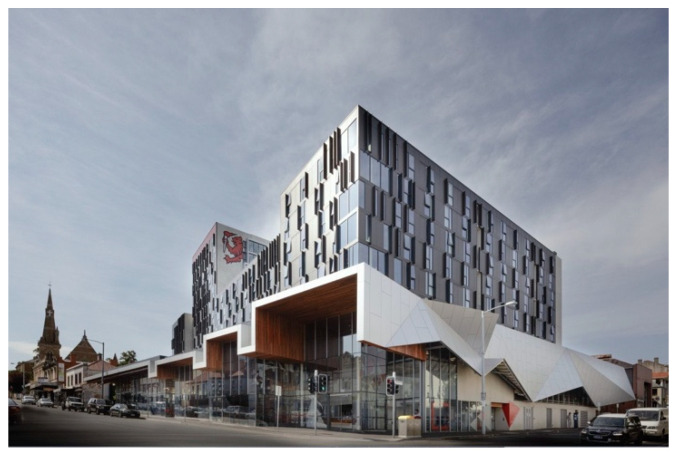
Street view of the apartment complex [Photographer: John Gollings AM].

**Figure 4 ijerph-17-07476-f004:**
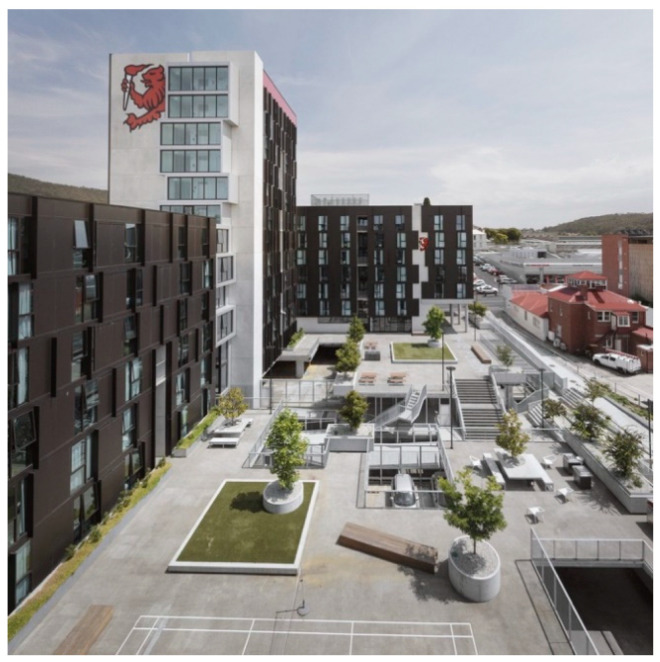
Aerial view of the proposed garden site and existing greenery in private and public areas. [Photographer: John Gollings AM].

**Figure 5 ijerph-17-07476-f005:**
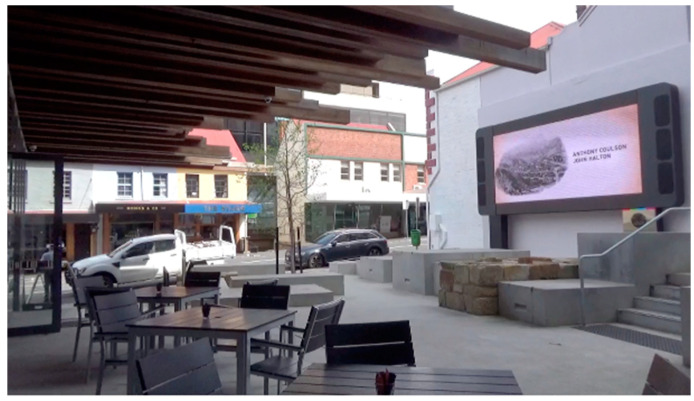
Café exterior, including “The Loop” video installation.

**Figure 6 ijerph-17-07476-f006:**
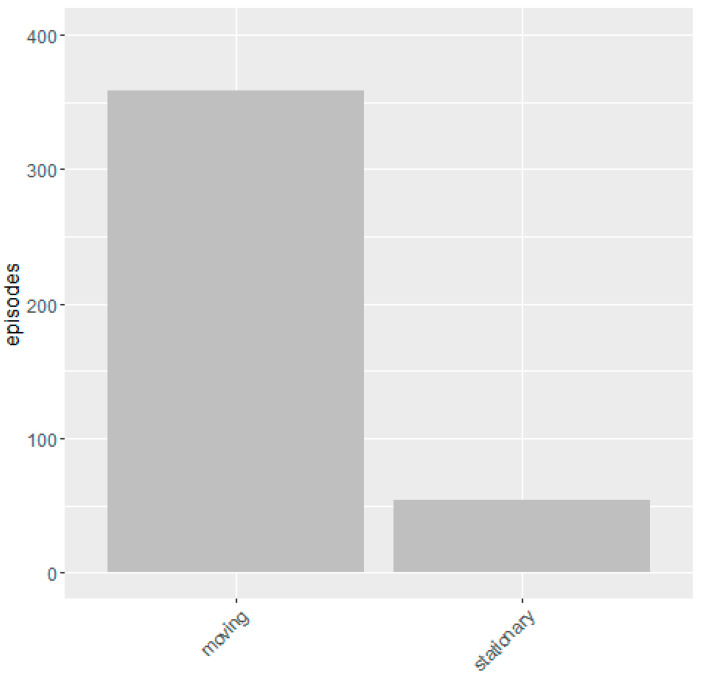
Episodes of movement and stationary activity observed.

**Figure 7 ijerph-17-07476-f007:**
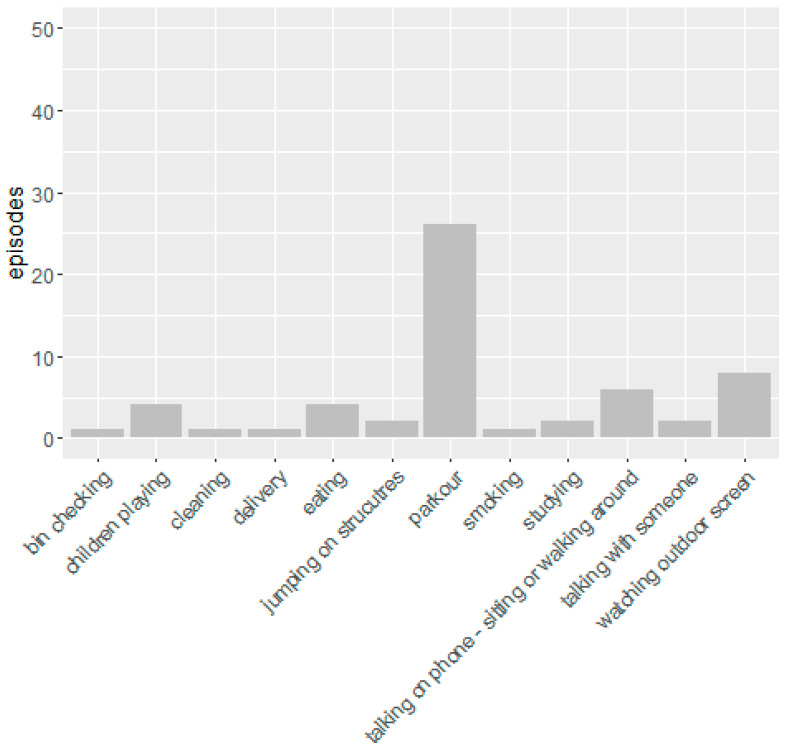
Types of activities observed.

**Figure 8 ijerph-17-07476-f008:**
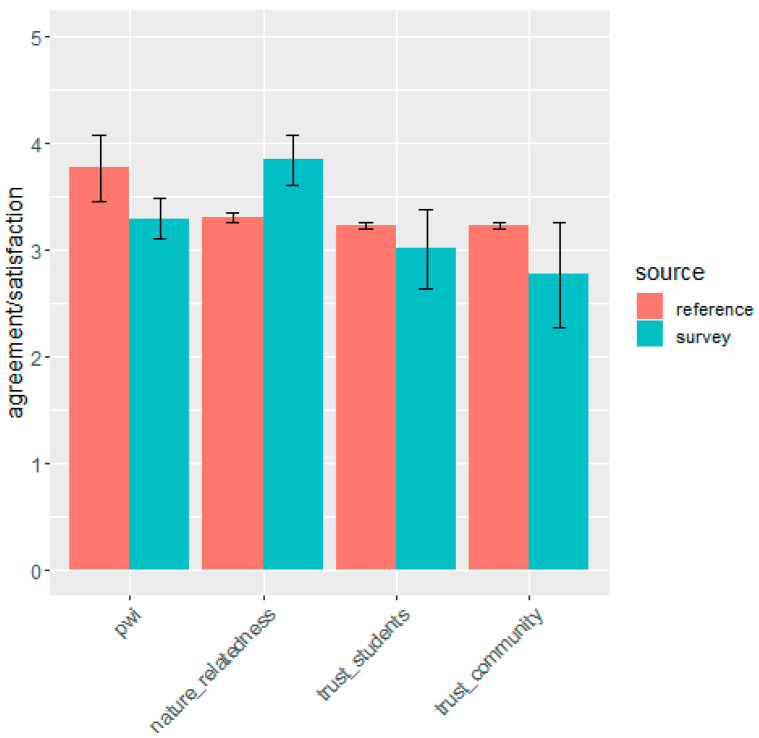
A comparison of respondents Personal Wellbeing Index (pwi), Nature Relatedness, and Trust in the university to engage with (a) students and (b) the local community, compared with reference results from other studies. 95% confidence intervals are shown.

**Figure 9 ijerph-17-07476-f009:**
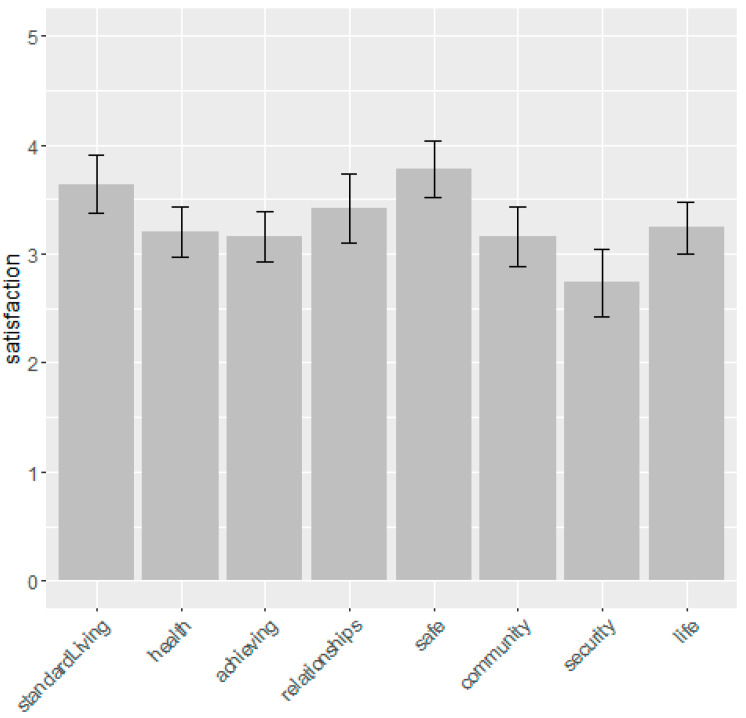
Respondents individual Personal Wellbeing Index scores. 95% confidence intervals are shown.

**Figure 10 ijerph-17-07476-f010:**
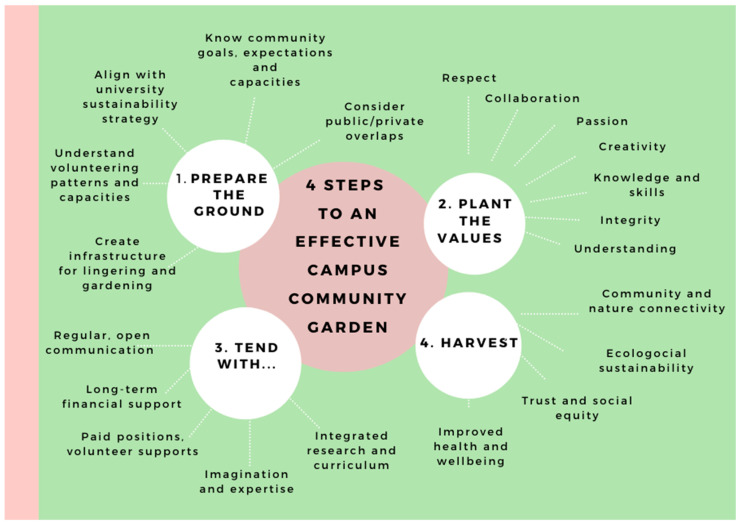
Four steps to an effective campus community garden.

**Table 1 ijerph-17-07476-t001:** Study methods with the associated data collection process and relevant stakeholders.

Method	Data Collection Detail	Stakeholders (Number)/Code
Interviews	Semi-structured, in depth, face-to-face	Local Business employees and owners (15 sites)/BS
Interviews	Semi-structured, brief, face-to-face	Members of the public utilizing the public outdoor space (*n* = 20)/PB
Focus Group	2-hour semi-structured discussion	Urban green practitioners and community gardeners (*n* = 10)/FG
Observation	Public Life observation (7 hrs, various time periods, 4 days) Video Footage (14 hrs, various time periods, 4 days)	Members of the public utilizing the spaces, including student residents
Survey	Online	Local communities, mostly focused on students residing in UTAS city accommodation (*n* = 64)

**Table 2 ijerph-17-07476-t002:** Observational Data summary.

Day/Time	Method/Period of Observation (min)	Weather Warm vs. Cool *	Episodes of Movement through (*n*)	Episodes of Stationary Activity (*n*)	Total Episodes of Public Life (*n*)
Thursday 9 a.m.	Researcher/40	cool	12	0	12
Thursday 1 p.m.	Researcher/40	warm	17	3	20
Thursday 4:05 p.m.	Research/40	cool	16	0	16
Thursday 7 p.m.	Researcher/40	cool	4	0	4
Friday 10 a.m.	Video/60	cool	39	1	40
Friday 12 noon	Video/60	cool	61	7	68
Friday 2:45 p.m.	Video/60	warm	42	4	46
Friday 4:20 p.m.	Video/60	cool	44	2	46
Saturday (1) * 10 a.m.	Researcher/40	cool	17	1	18
Saturday (1) 12 noon	Researcher/40	warm	15	1	16
Monday 9 a.m.	Video/60	warm	37	1	38
Monday 1 p.m.	Video/60	warm	54	5	59
Monday 3:30 p.m.	Video/60	warm	36	1	37
Saturday (2) 1 p.m.	Researcher/40	warm	20	3	23
Saturday (2) 3 p.m.	Researcher/40	warm	2	27	29
TOTALS	740 min		416	56	472

* The Saturday sessions were split over two to accommodate a major sporting event; Spring weather conditions were categorised into two broad descriptors by the researchers on site.

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
