# Peer review of "Trust, Connection and Equity: Can Understanding Context Help to Establish Successful Campus Community Gardens?"

_ijerph, 2020, doi:10.3390/ijerph17207476_

Round 1
Reviewer 1 Report
This is a refreshing and insightful empirical-conceptual study of an important but under researched socio-ecological problem of environmental health sustainabilities.
I recommend it be published 'as is'. Well done!
I wonder, however, if reporting on the study for publication is somewhat premature given the additional research questions recommended in the conclusion that, might I suggest, point to the 'lack' of a political-social theory 'driving' the way each of the mixed methods data analyses proceed 'independently' and are less amenable to a more dynamic, syncretic and stratified approach to inquiry, critique, and interpretation.
To be sure, there is much valuable insight and information contained in the study - some at a 'micro' level of key agency/agents, some at a 'meso' level of local interactions/players and community/business structures, including the University, some at a 'macro' level of cultural-sociological (and indication of relevant Tasmanian histories/demographies/geographies, but less so on politics for which Tasmania has a notorious 'split' of progressive and conservative 'forces') and some at the 'planetary' level of human-environment and culture-nature relations, actions, inactions over times-space). But making some additional political-social-ecological sense of what Bludhorn identified as the unsustainable politics of 'sustainability' (sic), despite good intentions and high expectations highlights the urgent need (for this reviewer) of a syncretic political theory to 'reassemble' and reinterpret (more holistically) the treatments of the various methods - the mixed need to be unmixed a bit, notwithstanding the limitations of a more stratified and layered politics of interpretation and representation. There are some useful approaches already available in social, political, and geographical/ecological type approaches/assemblages to inquiry and representation, particularly in the 'critical realist' literature (for example, Archer, James, Schatzki, Harvey, Giddens over the past three decades) and embodied-materialist 'ecofeminist' literature (for example, Salleh, Fremaux). All in different ways address the ontology, axiology, epistemology (including methodology) 'tensions' informing the needed incorporation of a 'political' theory (substantive in relation to the macro/planetary 'layer' but also the underlying assumptions revealed (in this small scale study) via the use of mixed methods.
While it probably is not important to 'rehearse' the above, the selection and adroit use of a supporting 'ecopolitical' theory will, I suspect, make it far easier to 'assemble' a syncretic and heuristic reinterpretation of what has been 'found' in this study which, in turn, will inform the redevelopment of future research questions.
The article is well constructed, nicely focussed and appropriately presented. Perhaps more could have been made about the 'health' benefits (physical, mental/emotionally, in particular, but also social - but also what 'ecocentrically' is 'in it' for nature, not just anthropocentrically?). More might have also been made of the vast literature on 'experiential' learning and education, following Dewey, and extending into early years education and children's outdoor/nature 'schools' play.
Amongst, other relevant or possible informants I have found useful, but of a philosophical type, see
Cooper, D. (2006). A philosophy of gardens. Clarendon Press.
Most of all, good luck and best wishes in the future development of this overdue and worthy project.
Reviewer 2 Report
This article discusses the potential for campus community gardens to improve the health and wellbeing of students and drive social equity. The research exemplifies the need for access to green space and areas for community connection among students. This article underlines the importance of understanding the context in which the CCG is going to be situated to improve its chances of success. Additionally, the research provides evidence-based insights on the type of community-oriented engagement needed throughout the CCG implementation/building process.
The article is well-organized and provides a clear overall understanding of the research intention, methodology, results, and proposed next steps. The research is particularly effective with its thorough mixed-methods approach and the wide range of voices captured. I appreciated the discussion pertaining to both the possibility for wellness from CCGs, as well as the anticipated regional and place-based constraints. Also, figure 10 is a unique and very appealing way of presenting the overall study recommendations.
Please note the following:
- Lines 104, 152, 186, and 189 need some editing
- The article would be improved by doing an overall copy edit
- In lines 320-322 you refer to Figure 8, but it seems like you are actually talking about Figure 9
Overall, this is a thoroughly researched piece of work and is, in my view, ready for publication upon receiving a final copy edit. I can see this work will be important for urban planners, university administrators, and various community members as it demonstrates the value of connection to the environment for mental wellbeing, and provides context into necessary considerations when creating these community spaces.
Reviewer 3 Report
The authors describe an important and so far under-researched area of community gardens in campuses of educational institutions.
The methods have been designed, used and discussed correctly - the Non-participatory on-site observation deserves special attention. The authors present possible distortions of the designs and discuss negative side effects, also referring to the future and current limitations related to the pandemic.
In addition to the first section (Introduction), it is worth mentioning the links between urban gardens and slow life values and urban lifestyles (Bartłomiejski, Kowalewski 2019), and urban resillience strategies in times of crisis (Hou 2020)
It is worth to consider more questions in the conclusions section regarding following issues: (1) the relationship between the popularity of the educational offer and facilities for students such as CCGs (2) o the place of ecology in PR of public education institutions
